

**Understanding of Morphometric Features for an Adequate Water Resources Management**
**in Arid Environments**
**Mohamed Elhag*[1], Hanaa K. Galal[2,3], and  Haneen Alsubaie[2]**
[1]Department of Hydrology and Water Resources Management, Faculty of Meteorology,
Environment & Arid Land Agriculture, King Abdulaziz University
Jeddah, 21589. Saudi Arabia.
[2]Biological Sciences Department, Faculty of Science, King Abdulaziz University,
Jeddah 21589. Saudi Arabia.
[3]Botany Department, Faculty of Science, Assiut University, Asyut, Egypt
*Correspondence email: melhag@kau.edu.sa*
**Abstract**
The current research investigates the significance of remote sensing data to evaluate the
fluviological features and to identify the morphometric parameters of Yalamlam basin, West of
Saudi Arabia. Hydrological characteristics, such as topographic parameters, drainage attributes,
and the land use, land cover pattern were evaluated for the water resource management of the
watershed area.  Under GIS environment, the delineation of the watershed and the calculation of
morphometric characteristics using ASTER GGDEM was undertaken. The drainage density of the
basin has been estimated to be very high which indicates that the watershed possesses high
permeable soils and low to medium relief (Hadley and Schumn 1961).  The stream order of the
area ranges from first to sixth order showing semi dendritic and radial drainage pattern that
indicates the heterogeneity in textural characteristics and is influenced by structural characteristics
in the study area. The bifurcation ratio (Rb) of the basin ranges from 2.0 to 4.42 and the mean
bifurcation ratio is 3.84 of the entire study area that signifies that the drainage pattern of the entire
basin is much more controlled by the lithological and geological structure. The elongation ratio is
0.14, which indicates that the shape of the basin belongs to the narrow and elongated shape. Land



use and land cover map were generated by using Landsat-8 image acquired on August 10th 2015
was classified to distinguish mainly the alluvial deposit from the mountainous rock.   The study
reveals that hydrological evaluation using ASTER GGDEM is more precise and applied with
compared to other techniques at the watershed scale.
**Keywords**: Hydrological characteristics, Morphometric characteristics, Remote Sensing and GIS
techniques, Wadi Yalamlam, Watershed management.
**1. Introduction**
The choice of an optimal interpolation method, for the prediction of soil properties at unsampled
locations, is a subject of great importance in agricultural management studies. Several attempts
had been made in order to specify the most accurate interpolation technique for the generation of
continuous soil attributes surfaces.
Most of the soil studies using interpolation techniques include the Inverse Distance Weighted and
Kriging methods. The accuracy of these methods has been compared in several studies. Gotway et
al. (1996) found that the IDW method generated more accurate results for mapping soil organic
matter and soil $NO_3$ levels. Wollenhaupt et al. (1994) compared these two interpolation techniques
and concluded that IDW was more accurate for mapping P and K levels of soil. Mueller et al.
(2004) observed that for the optimal parameters of the method, the accuracy of IDW interpolation
generally equaled or exceeded the accuracy of kriging at each scale of measurement. On the other
hand, other researchers observed Kriging to be more accurate for the interpolation of soil attributes.
Leenaers et al. (1990) found Kriging interpolation method the most accurate in comparison to
IDW, for mapping soil Zn content.



Other studies compared Kriging IDW and Radial Basis Functions interpolation techniques in soil
science. Schloeder et al. (2001) observed that Ordinary kriging and inverse-distance weighting
were similarly accurate and effective methods, while thin-plate smoothing spline with tensions
was not. Weller et al. (2007) resolved that not only the predications for kriging were not satisfied
the kriging method, but also was as good as any other radial base function interpolation.
Since the early decades of the century, spatial variability of soil properties has been widely studied
in order to understand the behavior of soils for agricultural purposes. Knowledge, of spatial
variability and relationships among soil properties, is important for the generation of soil maps for
reasons of land management.
Hydrological parameters are essential for adequate water resources management plans.
Morphometric characteristics aim to investigate the watershed delineation, site selection in water
recharge and discharge, run off modelling and other geomorphological studies (Sreedevi et al.,
2013; Elhag, 2015). GIS helps a wide variety of basin characterization and evaluation applications
under different terrain conditions (Pankaj and Kumar, 2009; Magesh et al., 2011).
Digital Elevation Model (DEM), such as ASTER GDEM (USGS, USA) is the keystone
involvement in various extractions of Geo hydrological parameters of the watershed. Several
parameters including slope, aspect, stream network, and upstream flow areas can be conducted
from the DEM characterization (Grohmann et al., 2007; Elhag, 2015). Reliable results of
implementing Remote sensing and GIS based morphometric evaluation using ASTER GDEM data
have been reported in numerous scholarly work of watershed characterization (Farr and Kobrick,
2000; Panhalkar, 2014; Elhag, 2015). Author's refers that ASTEDR GDEM is more useful and




very accurate tool for the watershed delineation and morphometric evaluation for the watershed
management.
The conducted results of the current study discuss that the analysis of methical information and
further hydrological and morphometric investigation can find out satisfactory alternative strategies
for adequate rainwater harvesting based on observed calculations in the designated study
watershed. The main aim of the present study is to identify and investigate various drainage
attributes to geometrical evaluation of Yalamlam basin for the sustainable rainwater harvesting
management and conservation of water resources.
**2. Materials and Methods**
**2.1. Study Area**
Wadi Yalamlam basin  is located about 125 km southeast of Jeddah city and is bounded by latitudes
20° 26′ and 21° 8′N and longitudes 39° 45′ and 40° 29′E (Figure 1). Wadi Yalamlam basin drained
large catchment area of about 180,000 hr. The basin boundary of the lower part is enlarged to
include nearly all the flat area in the downstream part. Wadi Yalamlam basin is initiated from high
elevation Hijaz escarpment with mean annual rainfall of about 140 mm. The basin elevations are
greatly varies from upstream and downstream parts and range between 2850 m and 25 m (a.s.l.)
respectively. The main course of Wadi Yalamlam is crosscut the highly fractured granitoids,
gabbroic and metamorphic rocks until the coastal plain of the Red Sea. The upper and middle parts
of Wadi Yalamlam basin are covered by intense natural vegetation. The lower part is covered
mainly by Quaternary deposits and sand dunes with small-scattered highly altered granitoids and
metamorphosed basaltic hills. Several basic dykes are recorded in the lower part of Wadi
Yalamlam basin. The thickness of Quaternary wadi deposits increased in the lower part (Elhag and





Bahrawi 2017). Regional groundwater flow drains toward the south and southwest following the
general trend of the main wadi channel. The gradient of the water table varies from one area to
another according to the variations in the pumping rates and hydraulic properties of the aquifer. It
has an average value of about 0.011. The high pressure of the subtropical zone in addition to local
topography affects climate in Yalamlam basin. Both regional and local circulations have a
dominant influence on the climate of the region. From the temperature records in the Red Sea coast
stations, the mean monthly maximum temperature is 38ºC and the mean monthly minimum
temperature is 20ºC. The highest recorded temperature in July is 49ºC and the lowest in January is
12ºC. The maximum mean monthly evaporation value is around 500 mm in summer, where in
winter         it         is         about         200         mm         (Elhag,         2016)

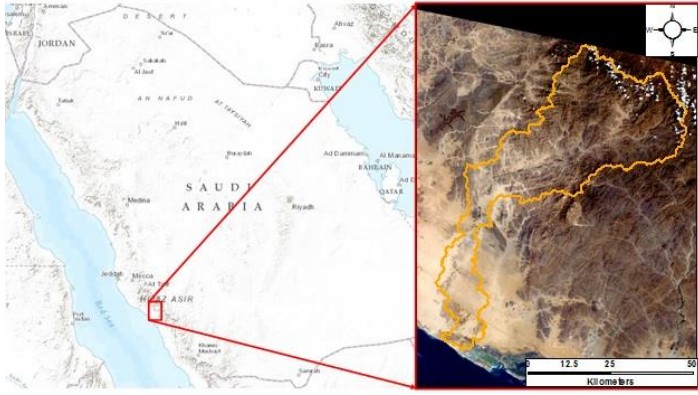


**Figure 1. Location of the study area (Elhag 2016).**
**2.2. Soil sampling**
Map accuracy and quality depends on the sampling method to be used, scale, analytical laboratory
errors and prediction errors. Sampling approaches depend on the objectives of the study that are
highly correlated to scale. Random Stratified Sampling was the adopted sampling design, the



landscape is divided into smaller areas, named strata, and afterwards 150 random samples are taken
from the designated study area.
**2.3. Physical and Chemical soil analysis**
Each individual sample was analyzed separately and each measurement was repeated three times
for the same extract. Thus, the final values of the measured attributes are represented by the mean
value of nine measurements. Soil samples were analyzed in order to estimate physical analysis
(clay, silt and sand) and chemical analyses (pH, Calcium Carbonate, Electric Conductivity,
Minerals and Organic Content) including all samples.
For standard particle size measurement, the soil fraction that passes a 2-mm sieve is considered.
Laboratory procedures normally estimate percentage of sand (0.05 - 2.0 mm), silt (0.002 - 0.05
mm), and clay (<0.002 mm) fractions in soils. Soil particles are usually cemented together by
organic matter; this has to be removed by $H_2O_2$ treatment. However, if substantial amounts of
$CaCO_3$ are present, actual percentages of sand, silt or clay can only be determined by prior
dissolution of the $CaCO_3$.
The chemical procedures presented here are extensive, though by no means exhaustive. For any
given element, there are several procedures or variations of procedures can be found in the
literature (Walsh and Beaton, 1973; Westerman, 1990). Procedure for determining soil pH in a 1:1
(soil: water) suspension was after McLean (1982). The methodology of EC measurement is given
in USDA Handbook 60 (Richards, 1954). Active $CaCO_3$ is usually related to total $CaCO_3$ equivalent,
being about 50% or so of the total value. Total $CaCO_3$, is currently estimated after Drouineau (1942).

**2.4. Interpolation techniques**





Geostatistics interpolation is based on the assumption that all values of a variable that is measured
are the result of a random process. The phrase "random process" does not indicate that all events
are independent. More specifically Geostatistics is based on random processes with dependence,
otherwise called autocorrelation, and relies on some notion of replication. Repeated observations
in nature can result in understanding the variation and uncertainty of natural phenomena, and
furthermore in estimating their sequence in space and time. Three interpolation techniques were
used for the generation of the prediction maps (interpolation). Inverse Distance Weighted (IDW),
Radial Basis Function (RBF) and Ordinary Kriging (OK) method were compared according to the
accuracy of the results.
Spatial distribution equation (Weibel, 1997):
$\gamma_{(k)=\frac{1}{2\,x\,n(k)}\,x\,\sum_{i=1}^{n(k)}[Z(X_i)-Z(X_{i+k})]^2}$
Where: $n(k)$ - number of pairs of observation; $Z(x_i)$ - soil property measured in point $x$, and in point
$x + k$.
Interpolation equation (Stoer and Bulirsch,, 1980)
$Z\,x\,(X_o) = \sum_{i=1}^{n}\lambda_i\,x\,Z(x_i)$
Where: $Z\,x\,(x_o)$ - interpolated value of variable $Z$ at location $X_o$, $Z(x_i)$ - values measured at location
xi, λi; - weighed coefficients calculated based on the semivariogram.
Trend and random error equation (Johnson and Riess, 1982)
$Z(S) = \mu(S) + \varepsilon(S)$
The symbol s stands for the location of the prediction location. $Z(S)$ is the variable you are
predicting (total extractable heavy metal concentration). $\mu(S)$ is the deterministic trend. $\varepsilon(S)$ is the
spatially-autocorrelated random error.
**2.5. Morphometric parameters**





Based on the foundation scholarly work of Horton (1945), Schumm (1963), Strahler (1964) and
others, several morphometric parameters were performed and computed utilizing ASTER GDEM
under GIS environment. Consequently, watershed delineation, stream network identification,
drainage frequency, drainage density, shape, elongation ratio, circularity ratio and form factor were
computed and evaluated using 30m spatial resolution of ASTER GDEM. The methodologies
adopted for the evaluation and computation of morphometric features are given in Table 1.
**2.6. Supervised classification**
Remote sensing data was obtained from Landsat Operational Land Imager (OLI-8) which is
acquired on June 10[th], 2013. Typical atmospheric and radiometric corrections and spatial
resolution enhancement were implemented for each band individually. Furthermore, supervised
classification was implemented using Support Vector Machine (SVM) classifier for better
classification results (Psilovikos and Elhag, 2013). The final step in the digital image analysis is
the evaluation of the accuracy of the computer derived classification results. These results are often
expressed in tabular form, known as a confusion matrix (Elhag et al., 2013). The SVM classifier
is implemented as:
$K\left(x_i, x_j\right) = tanh\left(g x_i^T x_j + r\right)$
Where:
$g$ is the gamma term in the kernel function for all kernel types except linear
$r$ is the bias term in the kernel function for the polynomial and sigmoid kernels.

**Table 1. Summary of the implemented morphometric features:**

| Item | Morphometric feature | Equation | Citation |
|---|---|---|---|
| 1 | Stream length ($L_u$) | Length of the stream | Horton (1945) |
| 2 | Stream length ratio ($R_L$) | $R_L = L_u/(L_u + 1)$ | Horton (1945) |




| 3 | Form factor ($F_f$) | $F_f = A/L^2$ | Horton (1945) |
| 4 | Drainage frequency ($F_d$) | $F_d = N_u / A$ | Horton (1945) |
| 5 | Drainage density ($D_d$) | $D_d = L_u / A$ | Horton (1945) |
| 6 | Drainage texture ($T$) | $T = D_d * F_d$ | Smith (1950) |
| 7 | Bifurcation ratio ($R_b$) | $(R_b) = N_u/(N_u + 1)$ | Schumm (1956) |
| 8 | Elongation ratio ($R_e$) | $R_e = D/L$ | Schumm (1956) |
| 9 | Mean bifurcation ratio ($R_{bm}$) | $R_{bm}$= average of bifurcation ratios | Strahler (1957) |
| 10 | Relief ($R$) | $R = H-h$ | Hadley & Schumm (1961) |
| 11 | Relief ratio ($R_r$) | $R_r = R/L$ | Schumm (1963) |
| 12 | Stream order ($S_o$) | Hierarchical rank | Strahler (1964) |
| 13 | Stream no | Order wise no of streams | Strahler (1964) |
| 14 | Mean stream length ($L_{sm}$) | $L_{sm}= L_u/N_u$ | Strahler (1964) |
| 15 | Circularity ratio ($R_c$) | $R_c = 4\pi A/P^2$ | Strahler (1964) |

Where
$A$ =Area of the basin (km$^2$)
$D_d$ =Drainage density
$F_f$=Form factor
$F_s$ =Stream frequency
$L$ =Basin length (km)
$L_{sm}$ = Mean stream length
$L_u+1$= The total stream length of its next higher order u
$L_u$ = The total stream length of order u
$N_u+1$= Number of stream segments of the next higher order
$N_u$ = The no. of stream segments of order u
$P$ =Perimeter (km)
$R_b$ = Bifurcation Ratio
$R_c$ =Circularity ratio
$R_e$ =Elongation ratio
$RL$=Stream Length Ratio
$T$=Drainage texture
$\pi$=3.14


## 3. Results and Discussion

Quantitative evaluation of the watershed through the analysis of morphometric parameter can

provide significant information about the hydrological characteristics of rocks, which are exposed

within the basin. The nature of drainage of a basin reveals reliable information about the

permeability of the rocks and the yield of the basin.



The ASTER GDEM has been collected with 30 m resolution and the GDEM has been used to
generate watershed area, aspects and slope map of the basin. The drainage network of a basin
depends on endogenous forces and exogenous forces. Geology and precipitation pattern of the
study area. ASTER GDEM was used for preparing aspects, slope map of the basin. Areal, relief
and linear aspects of the basin were analyzed under GIS environment. Currently a geospatial
technology has been given more precise and accurate information through the evolution of
morphometric parameter for the assessment of drainage watershed. GIS technology and satellite
data have been used to formulate data on spatial deviations in drainage attributes. Therefore,
drainage parameter provides a significant insight into hydrological characteristics, which is more
important to develop the watershed management strategies (YanYun et al., 2014).
Evaluation of drainage characteristics and other morphometric parameter of Yalamlam basin has
been undertaken to calculate the parameter and built topology of the basin. Different type of Arial
and linear aspects and their characteristics has been calculated, these aspects are, basin area (A),
basin length (L), basin perimeter (P), bifurcation ratio (Rb), elongation ratio (Re), circularity ratio
(Rc), drainage frequency (Fd) and drainage density (Dd) etc.

### 3.1. Stream order ($S_o$) and stream no:

The lower Yalamlam basin encompasses through the basin mega fan, which is formed by ancient
and modern radial drainage pattern in the designated study area. The channel of this area is
characterized by higher sinuous, decreased widths and lesser discharge capacity as a numerous
traverse Paleo alluvial channels (Bahrawi et al., 2016). Therefore, the stream ordering of the study
area has been ranked based on Strahler (1964) method and demonstrated in Table 2.

**Table 2. Stream network order based on Strahler method.**




| Strahler | Cnt_Strahler | Rb | Nu-r | Rb * Nu-r | Sum_Length |
|---|---|---|---|---|---|
| 1 | 598 | | | | 872.847 |
| 2 | 135 | 4.42963 | 733 | 3246.919 | 452.488 |
| 3 | 22 | 6.13636 | 157 | 963.409 | 237.306 |
| 4 | 6 | 3.66667 | 28 | 102.667 | 112.047 |
| 5 | 2 | 3.00000 | 8 | 24.000 | 54.635 |
| 6 | 1 | 2.00000 | 3 | 6.000 | 58.259 |
| **Sum** | 764 | 19.232659932 | 929 | 4342.99427609428 | 1787.582302 |
| **Mean** | | **3.847** | | **4.675** | |


## 3.2. Bifurcation ratio:

The bifurcation ratio was calculated by the no of streams of an order to the no of the streams of
the next higher order. The values vary from 2.0 to 4.42 of the Yalamlam streams basin, which is
also signified the maximum structural influences (Strahlar, 1964). After the calculation of
bifurcation ratio, calculate the average value which is the mean bifurcation ratio is 3.84 of the
basins. The value also indicates that the drainage pattern has been affected by structural
disturbances within the basin. The obtained no of bifurcation ratio varies from one order to another
order. Such variation is interpreted as irregularities of the lithological and the geological
development within the watershed. The values of bifurcation ratio and mean bifurcation ratio have
been shown in Table 3.

## 3.4. Drainage texture and drainage density:

Drainage density is an expression of spacing and distribution of channels as proposed by Horton
(1932) that measure the total length of the streams of all orders as calculated with per unit area.
Relative relief and slope gradient of the river basin primarily control the stream density. The stream
density of the watershed has been calculated which is shown in Table 3. The value of drainage



density is 0.92 in the study basin. The drainage density has been classified into five kinds of
drainage texture as proposed by Smith (1950).
The drainage density more than 8 indicates very fine, the value 8-6 is fine, between 6-4 is moderate,
the value 4 to 2 is coarse and less than 2 represent very coarser drainage texture. The observer
drainage texture is 0.138, signifies the resistant permeable rock with moderate infiltration rate and
moderate relief (Bahrawi et al., 2016). The value of the variation of drainage texture depends on
different kind of natural factors that is rainfall and other climatic characteristics, rock, soil type,
vegetation characteristics, permeability, relief, infiltration capacity within the watershed. The
relationship between the hydrological features and the geological structures is estimated to be with
a high drainage density caused by the mountainous relief in the basin.  The lower value of drainage
density reveals that, the region is composed of permeable sub surface material, low relief, dense
vegetal cover which results in the increase of more infiltration capacity in the basin. The value of
high drainage density indicates mountainous relief, thin vegetation and impermeable sub surface
material, highly resistant rock types in the river basin.
**3.5. Drainage frequency:**
Drainage frequency or stream frequency is calculated by the total no of streams per unit area of all
stream orders that proposed by Horton (1932). The correlation value of drainage density and
stream frequency is plays positive of the basin, which suggests that the no of streams, population
has increases with the increase of drainage density. The observed value of stream, frequency is
about of 0.34 for the watershed exhibits the highly positive connection with stream density that
has been shown in Table 3.
**3.6. Elongation ratio:**



The elongation ratio is calculated by the ratio between maximum length of the basin and the
diameter of a circle, which fitted in the same basin area, as proposed by Schumm, (1956). The
elongation ratio value generally varies between 0.6 and 1.0 where a wide variety of geological
condition and climatic characteristics has. According to Strahlar (1964) the values close to 1.0
represent the region belongs to very low relief with less structural influences and the value ranges
from 0.8 to 0.6 are generally associated with much steep slope and high relief. The values of
elongation ratio can be categorized into 3 groups, namely less than 0.7 indicates elongated shape,
and 0.8 to 0.9 values represent the oval shape and more than 0.9 values represent the circular shape
of the basin. So, the elongated ratio of the study area is 0.14 that suggests that the basin shape is
belongs to the much more elongated type (Table 3) of the basin where structural influence is much
all over the basin.
**3.7. Circularity ratio:**
According to Miller (1953) circularity ratio is the ratio between the area of a circle, which fitted
in the basin perimeter, and the total basin area. Circularity ratio is much more influenced by
geological structure, relief, slope, climate, frequency and length of stream and land use land cover
within the basin. The basin circularity ratio is 0.08, which represents that the basin is strongly
elongated and belongs to the heterogeneous geological structure and materials. The observed
values also signify the high run off capacity and low permeable capacity of subsoil and sub-surface
soil along the basin area (Table 3).
**3.8. Form factor:**
Horton (1932) defines the form factor as the ratio between the square of the basin length and basin
area. The values of form factor represent the flow intensity of the study area. Generally, the


elongation shape and the values of form factor have a negative relationship that means the smaller
value indicates the more elongated shape of the basin. The values should always be not exceeds
0.7854, higher value of form factor represent the higher peak flows of a higher period. The
observed value of form factor is 0.06 of the Yalamlam watersheds signifies the elongated shape of
the basin (Table 3). Therefore, the lower values and elongated shaped basin indicates that the
watershed belongs to the flatter peak flow of shorter duration.
**Table 3. Wadi Yalamlam morphometric features**

| Parameters | Descriptions | Remarks |
|---|---|---|
| Basin area (km$^2$) | 1940.3 | The basin area is too large |
| Basin length (km) | 60.56 | Basin length is very high |
| Basin perimeter (km) | 417 | High basin perimeter |
| Elongation ratio | 0.14 | Elongated |
| Form factor | 0.06 | Elongated shape and flatter peak flow |
| Circularity ratio | 0.08 | Strongly elongated and heterogeneous geological structure |
| Drainage frequency | 0.34 | Low stream frequency |
| Drainage density | 0.92 | Drainage density is considerably high |
| Drainage texture | 0.138 | Highly resistant permeable rock with moderate infiltration rate |


### 3.9. Relief and relief ratio if the watershed:

Relative relief is the difference between the highest and lowest elevation of the watershed. The
relief ratio is the ratio between relative relief and the maximum length of the basin as proposed by
Schumm, (1956). It can analyses the steepness of the basin and evaluate the intensity of erosion
process of the study area. Here the relief ratio is 4.17, which indicated that most of the designated
basin is situated along the mountainous rough slope and much narrower in the lower areas.
**3.10. Slope map:**



Slope is the ratio between horizontal and vertical surface, of a region, which can be expressed by
the percentage and degree. It was found that the most of the area (upper middle part) of Yalamlam
watershed comes under steep, very steep and very high steep slopes that indicate the area having
a much mountainous topography. The main channel slope of the basin comes under gentle slope
(0.042) which designed to flat topography and excellent for the ground water management through
favoring for infiltration.
**4. Geo hydrological Inferences from Morphometric Evaluation:**
The classification of Remote Sensing data was to quantify the area of all alluvial deposits to the
bare rock area within the designated study area as illustrated in Figure 2. Table 4 indicates that the
area of the alluvial deposits is roughly equal to the bare rock area which means that the watershed
is the watershed is likely to be used for rainwater harvesting (Elhag, 2014 and Elhag and Bahrawi
2014 a).
**Table 4. Wadi Yalamlam land cover classifications**

| Land cover category | Area sq. km | Percentage (%) |
|---------------------|-------------|----------------|
| Vegetation          | 82.7        | 4.26           |
| Alluvial deposit    | 803.3       | 41.4           |
| Bare rocks          | 1054.3      | 54.3           |
| **Total**           | **1940.3**  | **100**        |





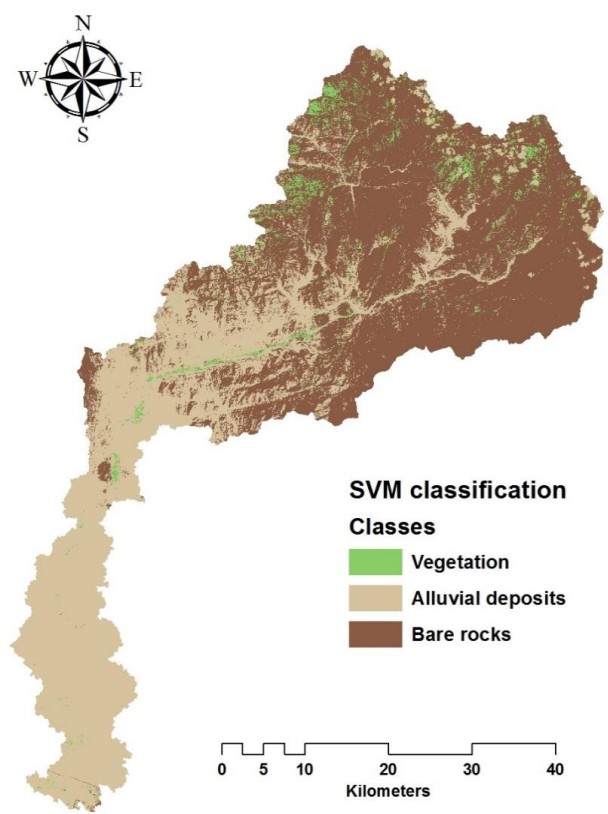


**Figure 2. Supervised classification of Yalamlam basin**.

Morphometric evaluation of the Yalamlam basin based on ASTER GDEM with remote sensing
and GIS techniques is the most significant method for proper and precise interpretation of
hydrological parameters of any terrain features. It is also indirectly influence the hydrological
status of the basin. The quantitative morphometric evaluation has an intense utility to watershed
delineation, water and soil conservation and their management for future sustainability. The
morphometric analysis of the Yalamlam basin shows that the watershed has narrow, elongated
shape and very high mountainous relief. The planning for runoff and artificial recharge of the area
has been selected based on small-scale topographical maps and low relief (gentle slope) in the



lower area of the basin. Also drainage morphometric makes a positive role through GIS and remote
sensing techniques by selecting artificial recharge sites and the creation of demand storage point
in the mountainous region with the basin. In addition, if the morphometric information is integrated
with other hydrological parameter of the river basin, the strategy for water harvesting and
recharging measures gives a better plan for groundwater management and development for the
future.
The drainage pattern of the basin is sub dendritic and radial in nature. The pattern was affected by
more or less heterogeneous structural and lithological characteristics. In addition, high drainage
density is observed all over watershed along with very high relief and impermeable subsoils and
land rock substratum, mountainous terrain slope. On the other hand, lower riparian areas are low
drainage density, which are favorable for the identification of water storage areas and ground water
potential zones. However, slope plays a significant role in determining the relation between
infiltration rate and runoff velocity where infiltration rate is inversely controlled by regional slope.
So all evaluated parameters are more important to the analysis of future water availability of the
study region.
Results obtained from pervious scholarly work of Şen, (1995) pointed out that the average
transmissivity values calculated within the study area range from 91 to 147m$^2$/day. While the the
transmissivity values increase sharply in the downstream area to range between 267 and 731
m$^2$/day (average 500 m$^2$/day). Such findings support the hypothesis that the aquifer is of high
potential therein. On the other hand, the hydraulic conductivity values calculated for Yalamlam
basin attain a high hydraulic conductivity (16 m/day) due to more permeable alluvial deposits (Sen,
1995, Elhag and Bahrawi 2014 b,c).



**5. Conclusions:**

The evaluation of hydrological characteristics of the Yalamlam watershed confirms that the area is having high relief and the shape is elongated in nature. The stream network of the watershed is basically dendritic type in lower portion which indicates the lack of structural influences and the homogeneity of textural characteristics, but the upper portion of the watershed highly influenced by tectonic and structural activity due to the parallel pattern of the drainage network. The drainage characteristics of the basin help to understand the different kind of terrain parameters, i.e, runoff, infiltration capacity and nature of the bedrock etc. The drainage density and frequency of the drainage basin is low that indicates the high permeability rate and well-drained capacity of the sub surface formation. All kinds of basic and derived parameters reveal an important the water recharge areas and measures can be undertaken for the soil conservation structures and water resource management. Therefore, watershed analysis using remote sensing data, GDEM data and GIS techniques has an efficient, precise tool for the understanding of any terrain attributes such as surface runoff, nature of bedrock, and infiltration capacity helps to better understanding of drainage evolution and management of ground water potential and status of landform and their characteristics for watershed management and planning. The study will be useful for water as well as natural resource management of any terrain at the watershed level. For sustainable water resource management and watershed development decision makers and planners can be used these kinds of hydrological analysis.

The thickness of the saturated zone within the aquifer varies from less than 1 m upstream of Yalamlam to about 30 m in the Sa'diyah area. The aquifer is generally unconfined, especially in the upper parts of the wadi. Semi-confining conditions may exist in the lower parts where layers



of clay exist. There are about 31 wells in the basin of Wadi Yalamlam, out of which, 23 are hand-
dug wells and the others are drilled.
**Acknowledgement**
This article was funded by the Deanship of Scientific Research (DSR) at King Abdulaziz
University, Jeddah. The authors, therefore, acknowledged with thanks DSR for technical and
financial support.

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
