# Peer review of "Understanding of Morphometric Features for an Adequate Water Resources Management in Arid Environments"

_Geoscientific Instrumentation, Methods and Data Systems, 2017_

## Referee Comment (RC1) · S. Boteva (Referee) · 13 May 2017

Dear Editors, Dear Authors, In the following letter I will submit my comments as follows:

General comments The submitted article "Understanding of Morphometric Features for an Adequate Water Resources Management in Arid Environments" is in the scope of the journal and represents a very interesting approach for water resources management. It contains all the necessary parts for a scientific article. The authors clearly prove the benefits of using ASTER GGDEM not only for water but also for other natural recourses management, which is supported with detailed analysis. There are some sentences that should be rewritten for better understanding: - Page 1 "The drainage density of the basin has been found very high indicates...." - Page 10 "The variation interpreted that the irregularities of the lithological and geological development within the watershed" - Page 11 "The relation between hydrological characteristics and geological structures..." and "The correlation value of drainage density and stream frequency..." Specific comments Introduction I think there should be one or two more sentences linking the review information between soil properties and hydrological parameters. Materials and methods Part 2.3. Soil sampling – If a method for soil sampling is used it should be cited. Part 2.3. Physical and Chemical soil analysis - it would be better if the methods used for soil analyses are given. Conclusions The last two paragraphs of the conclusion are appropriate for example: the first for description of the site and the second - for the Results and discussion part.

Technical comments Comments concerning the English language: It should be written: - Page 3 Morphometric characteristics aim... - Page 3 The conducted results of the current study discuss that... - Page 5 Map accuracy and quality depend on... - Page 9 aspects of the basin were analysed... - Page 12 which represents that ... Sentences where verb is missing: It should be written for example: - Page 1 "characteristics and is also.. " - Page 3 "Several parameters including slope, aspect, stream network, and upstream flow areas Grohmann et al., 2007; Elhag, 2015)" - Page 7 "which is acquired on June... " - Page 7 "These results are often... " - Page 8 "of rocks which are exposed... " - Page 9 "which is formed by ancient... " - Page 9 "channel of this area is characterized... " Technical corrections: - Page 14 - citation (Elhag and Elhag) - not correct - All years in the References part should be given in brackets As a whole, I recommend this article to be published because of its high scientific and practical value. All my comments aim to improve its quality and to meet all the requirements of the journal.

Yours sincerely,

Silvena Boteva

28, 2017.

---

## Author Comment (AC1) · 13 Jun 2017

thank you, my dear reviewer, please rest assured that all of your valuable comments will be followed literally yours

---

## Referee Comment (RC3) · Anonymous Referee #3 · 2 Jul 2017

Reviewer Responses' to

"Understanding of Morphometric Features for an Adequate Water Resources Management in Arid Environments"

General comments The topic of the article Understanding of Morphometric Features for an Adequate Water Resources Management in Arid Environments is within the scope of Geo- Scientific Instrumentation, Methods and Data Systems Discussions. The title clearly reflects the contest of the paper. The aim of the research is to identify and investigate various drainage attributes to geometrical evaluation of Yalamlam basin for the sustainable rainwater harvesting management and conservation of wa-

ter resources, which will ease decision-making processes regarding sustainable water resources management. The introduction provides a generalized background of the topic, the variety of techniques used for morphometric features estimation. The Materials and methods section describes each step of the work methodology in detail along with the conducted calculations. Obtained results are sufficiently interpreted and discussed. The authors clearly describe their original contribution in estimating the morphometric features and especially its application in arid regions. The conclusions underline the benefits for the decision makers when the morphometric features are given. The article is well structured with all the necessary parts for such kind of a scientific work. The references are appropriate and up-to-date.

Specific comments There are some modifications needs to be considered to improve the quality of the work. These modifications are listed as comments to the attached file.

Please also note the supplement to this comment:
https://www.geosci-instrum-method-data-syst-discuss.net/gi-2017-28/gi-2017-28-RC3-supplement.pdf

**Supplement:**

**Understanding of Morphometric Features for an Adequate Water Resources Management**

**in Arid Environments**

**Mohamed Elhag*[1], Hanaa K. Galal[2,3], and  Haneen Alsubaie[2]**

[1]Department of Hydrology and Water Resources Management, Faculty of Meteorology,

Environment & Arid Land Agriculture, King Abdulaziz University

Jeddah, 21589. Saudi Arabia.

[2]Biological Sciences Department, Faculty of Science, King Abdulaziz University,

Jeddah 21589. Saudi Arabia.

[3]Botany Department, Faculty of Science, Assiut University, Asyut, Egypt

*Correspondence email: melhag@kau.edu.sa*

**Abstract**

The current research investigates the significance of remote sensing data to evaluate the fluviological features and to identify the morphometric parameters of Yalamlam basin, West of

Saudi Arabia. Hydrological characteristics, such as topographic parameters, drainage attributes, and the land use, land cover pattern were evaluated for the water resource management of the watershed area. Under GIS environment, the delineation of the watershed and the calculation of morphometric characteristics using ASTER GGDEM was undertaken. The drainage density of the basin has been estimated to be very high which indicates that the watershed possesses high permeable soils and low to medium relief (Hadley and Schumn 1961).  The stream order of the area ranges from first to sixth order showing semi dendritic and radial drainage pattern that indicates the heterogeneity in textural characteristics and is influenced by structural characteristics in the study area. The bifurcation ratio (Rb) of the basin ranges from 2.0 to 4.42 and the mean bifurcation ratio is 3.84 of the entire study area that signifies that the drainage pattern of the entire basin is much more controlled by the lithological and geological structure. The elongation ratio is

0.14, which indicates that the shape of the basin belongs to the narrow and elongated shape. Land

Commented [R.1]: Unnecessary text

Commented [R.2]: Are essential to evaluate

[revised manuscript text omitted]

**Commented [R.18]:** I think you mean three not nine

**Commented [R.19]:** analyses

**Commented [R.20]:** rephrase

**Commented [R.21]:** please omit

For standard particle size measurement, the soil fraction that passes a 2-mm sieve is considered.

Laboratory procedures normally estimate percentage of sand (0.05 - 2.0 mm), silt (0.002 - 0.05

mm), and clay (<0.002 mm) fractions in soils. Soil particles are usually cemented together by organic matter; this has to be removed by $H_2O_2$ treatment. However, if substantial amounts of

$CaCO_3$ are present, actual percentages of sand, silt or clay can only be determined by prior dissolution of the $CaCO_3$.

The chemical procedures presented here are extensive, though by no means exhaustive. For any given element, there are several procedures or variations of procedures can be found in the literature (Walsh and Beaton, 1973; Westerman, 1990). Procedure for determining soil pH in a 1:1

(soil: water) suspension was after McLean (1982). The methodology of EC measurement is given in USDA Handbook 60 (Richards, 1954). Active $CaCO_3$ is usually related to total $CaCO_3$ equivalent, being about 50% or so of the total value. Total $CaCO_3$, is currently estimated after Drouineau (1942).

**Commented [R.22]:** there are no need for any chemical analysis as its not taken any part of the results part of the discussion part

**2.4. Interpolation techniques**

[revised manuscript text omitted]